# Modern Approaches for the Genetic Improvement of Rice, Wheat and Maize for Abiotic Constraints-Related Traits: A Comparative Overview

Elena Benavente *[ID] and Estela Giménez *[ID]

Department of Biotechnology-Plant Biology, School of Agricultural, Food and Biosystems Engineering, Universidad Politécnica de Madrid, 28040 Madrid, Spain
* Correspondence: e.benavente@upm.es (E.B.); mariaestela.gimenez@upm.es (E.G.); Tel.: +34-910670837 (E.B.); +34-910670865 (E.G.)

**Abstract:** After a basic description of the different sets of genetic tools and genomic approaches most relevant for modern crop breeding (e. g., QTL mapping, GWAS and genomic selection; transcriptomics, qPCR and RNA-seq; transgenesis and gene editing), this review paper describes their history and the main achievements in rice, wheat and maize research, with a further focus on crop traits related to the improvement of plant responses to face major abiotic constrains, including nutritional limitations, drought and heat tolerance, and nitrogen-use efficiency (NUE). Remarkable differences have been evidenced regarding the timing and degree of development of some genetic approaches among these major crops. The underlying reasons related to their distinct genome complexity, are also considered. Based on bibliographic records, drought tolerance and related topics (i.e., water-use efficiency) are by far the most abundantly addressed by molecular tools among the breeding objectives considered. Heat tolerance is usually more relevant than NUE in rice and wheat, while the opposite is true for maize.

**Keywords:** genetic tools; genomics; crops; drought tolerance; heat tolerance; nitrogen-use efficiency; bibliometrics





## 1. Introduction

The genetic structure of plants has been manipulated by farmers since agriculture began 10,000 years ago. For centuries, the domestication process has progressed on the basis of selecting seeds of the grains or fruits from the individuals best adapted to local human needs and practices. This manipulation of natural diversity has outlined the genetic architecture of currently existing crops, which in some instances differ radically from their wild ancestors [1].

Plant improvement has been carried out throughout history using the scientific knowledge available. Therefore, soon after the discovery of plant sexual reproduction, early breeders implemented crossing designs to increase the variability available for selection. The rediscovery of Mendel's laws of inheritance and the spread of Darwin's ideas on the evolution of species in the early twentieth century provided a founding theoretical basis; thus, that crop improvement experienced a major transformation. The recognition of the value of genetic variability and the ability to predict the phenotypic outcome of designed crosses were a revolution that turned crop improvement into a scientific discipline in a few years [2].

Conventional genetic improvement, based on the application of classical genetic principles of trait transmission, has successfully introduced suitable characteristics into cultivars. During the breeding process, offspring must be tested in each generation to select superior individuals as parents of the next generation until the desired combination of crop traits is found.

The classical breeding approach has two main limitations: (i) only genetic variability existing in the crop species itself or in cross-compatible species can be used for crop improvement; (ii) the selection of superior genotypes is based on phenotypes. This may significantly affect the investment to develop new varieties; for instance, if trait evaluation requires time-consuming methods or expensive equipment, or when a large number of plants needs to be grown until maturity in field conditions. Moreover, for some complex polygenic traits, individual's phenotypes can be unreliable, and the identification of valuable genotypes requires phenotyping of their progenies (i.e., progeny testing) (Chapters 5 and 8 in ref [3]).

Modern breeding has addressed these limitations by incorporating biotechnological tools that can overcome cross-sexual barriers or create novel variability in the crop species, or that are ultimately applied to improve the reliability of selection programs based on the genotyping of segregant progenies [2]. The development of advanced tools for the qualitative and quantitative analysis of gene expression in organs and tissues must also be highlighted, because these methodologies provide the molecular basis to establish functional links between genes and traits [4]. Some of the earliest methods, introduced by the 1980s, have been virtually replaced by new techniques or approaches that allow the addressing of similar goals either with increased efficiency, by saving time, or both. The most recent incorporations to the molecular breeding toolbox have emerged from the current capacity to sequence organisms at the genome level combined with bioinformatics developments. The genome contains all the individual's hereditary information and Genomics has become essential to understand how genes control the traits exhibited by an organism and how the traits are transmitted to its offspring. Therefore, plant genomics has enormous potential, providing very valuable information that can be exploited for varietal improvement in breeding programs [5].

Rice, wheat, and maize are acknowledged as the most relevant crops worldwide, not only because of their leading position in agricultural and economic terms (Figure 1), but also because these three crops represent more than 50% of the human daily caloric intake. It can be added that maize has also a major indirect role in human nutrition by supplying most of the nutrients for livestock, along with other cereal grains (i.e., barley and sorghum) [6].

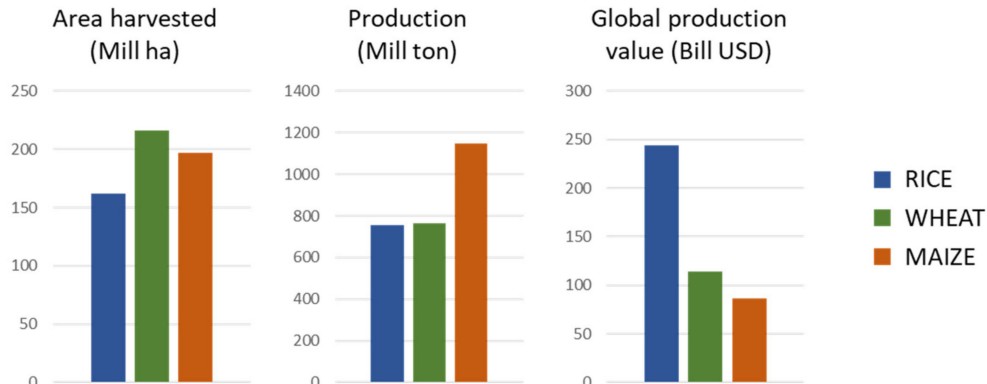

**Figure 1.** Global data illustrating the agricultural and economic relevance of rice, wheat, and maize. Area harvested and production correspond to 2019, while global gross production value corresponds to 2018. (Source: FAOSTAT database, in www.fao.org/faostat, accessed on 27 January 2021).

Some distinctive biological features of rice, wheat and maize must be taken into account to understand the remarkable differences regarding the timing and usage of some biotechnological developments in each of these crops (see below), in spite of a similar interest which can be assumed to exist on the breeders' side.

Rice (*Oryza sativa* L.) is a diploid species ($2n = 2x = 24$) with the smallest-sized genome among all the crop plants of economic importance (430 Mbp). It explains the earliness of its full genome sequencing compared to any other main crops [7,8]. Other advantages, such

as the availability of easier and efficient transformation protocols by the early 1990s, led to the adoption of rice as a model cereal for molecular studies [9]. Accordingly, all fine mapping and reverse-genetic tools, first developed in the model plant *Arabidopsis thaliana*, were soon applied to the structural and functional analysis of rice genes. Rice is likely the crop plant with the largest genomic toolbox [10–13]. Its panel of analytical resources also includes several wide collections of germplasm resources, most of which were developed and are being used by international consortia [14,15].

Bread wheat (*Triticum aestivum* L.) is a hexaploid species ($2n = 6x = 42$ chromosomes) with one of the largest genomes among all crop plants (around 16,000 Mbp). The wheat genome is composed of three homoeologous sub-genomes, with a basic number of $x = 7$ each (AABBDD genome composition). Its large and structurally complex genome has hindered the application of plant molecular tools to wheat and explains why the sequencing of the wheat genome has not been completed until very recently [16]. Notwithstanding, the work of Sears in the 1940–1950s [17,18] made a stock of more than 250 aneuploid lines of the wheat cultivar Chinese Spring available, which has been used for the mapping and expression studies of many wheat genes, even before the advent of DNA-based analytical tools [19]. For the same purposes, chromosome-engineered deletion, substitution, and recombinant lines derived from the initial Sears' lines, or created from other wheat cultivars, have also been (and are still being) extensively employed [20]. Durum or pasta wheat (*Triticum turgidum* L.; $2n = 4x = 28$, AABB genome composition) is not as relevant as bread wheat from an economical point of view, although has often been pioneering in the application of DNA-based methodologies to commercial wheat crops because of its less complex genetic/genomic architecture (tetraploid versus hexaploid). Although the genome of the reference cultivar Svevo was not published until 2019 [21], previous assemblies of the durum wheat genome were key to complete the sequencing of the bread wheat genome [22].

Maize (*Zea mays* L.; $2n = 2x = 20$) has a medium-sized genome of about 2300 Mbp (in the range of the human genome) that was sequenced in 2009 [23]. In spite of its strictly diploid behavior, there is much molecular evidence of its ancient polyploid origin [24,25]. From a breeder's perspective, maize is a crop radically different from rice and wheat. The distinct mode of reproduction (allogamy in maize and autogamy in rice and wheat) is on the basis of the different classical breeding methods that can be applied in each case [3]. Maize was the first crop where the superiority of hybrids over their parental lines, a biological phenomenon known as heterosis, was discovered and exploited for plant improvement; hence, most current elite maize varieties are F1 hybrids. Rice and wheat varieties are mostly inbred pure-lines that can be easily multiplied in the farm by the naturally occurring autogamous reproduction of these self-pollinated species, However, hybrid genotypes can only be multiplied by the parental lines' owner, and hybrid seed must be bought every year. This makes maize varieties the most profitable for seed companies and cost-effective for expensive biotechnology investment.

This review describes the most common breeding-assistant tools that are based on the molecular knowledge of genes and genomes, grouped in three categories according to their primary purpose: marker-assisted breeding, gene expression analysis, and genetic modification. We have aimed to provide a comparative overview of their timing and usage in rice, wheat, and maize, with a special focus on milestones and specific achievements regarding the improvement of crop attributes involved in plant response to major climatic constraints and nutritional limitations—drought and heat tolerance and nitrogen-use efficiency. These three goals are the main crop breeding objectives to overcome the most challenging abiotic threats to global food security [26]. With that in mind, we have conducted: (i) a bibliographic search on the first documented evidence of use of the main molecular tools in these major crops; and (ii) a bibliometric analysis on the number of records retrieved from the Web of Science database for specific tools, globally and locally, for the breeding topics of interest. Details on the search queries and the methodology followed for these analyses are provided as Supplementary Materials.

## 2. Marker-Assisted Breeding

A molecular marker is any polymorphism at the DNA level. DNA-based markers are much more numerous (theoretically, up to one marker per base-pair of the crop genome) and more evenly distributed throughout the whole genome than classical genetic markers (i.e., morphological or biochemical). Other practical advantages in a crop breeding context are that their presence or absence is not affected by the environment and that can be detected at any stage of plant growth.

The term marker-assisted breeding (MAB) is used to describe several modern breeding strategies that are based on using the molecular marker profiling determined in segregant progenies or populations as the criterion to select the superior individuals for the trait of interest. The different MAB approaches can be classified into two main groups, marker-assisted selection (MAS) and genomic selection (GS), which mainly differ in the number of markers used for a genotype-based selection.

### 2.1. Marker-Assisted Selection

Marker-assisted selection, or MAS, is the most fruitful practical application of biotechnological tools for crop breeding to date. To be useful for selection, a marker must be reliably associated to the phenotype of interest. An ideal DNA marker is located within the sequence of the gene controlling the target trait, allowing the discrimination of favorable from unfavorable alleles. However, it requires previous structural and functional gene characterization, which is frequently lacking, especially regarding polygenic quantitative traits. Nevertheless, a molecular marker closely linked to the target gene or quantitative trait locus (QTL) may serve to conduct a preliminary selection of promising genotypes in large segregating populations or at early plant developmental stages, thus making MAS a cost-saving strategy.

Some pre-requisites must be fulfilled for the successful implementation of a marker-assisted breeding strategy [27]:

(a)    A suitable marker system. The relatively recent development of high-throughput Next Generation Sequencing (NGS) platforms has revolutionized the field, enabling and generalizing the use of single nucleotide polymorphisms (SNPs) as molecular markers. Earlier marker systems not based on NGS methods had several disadvantages related to expensiveness, time-consumption, or reproducibility.

(b)    The development of genetic or physical maps, where the marker–trait associations can be contextualized at a genome level and the most suited markers can be chosen for MAS. High-density genetic linkage maps, based on the segregation of markers and genes in experimental populations, have been built in most economically important plant species for MAS applications. However, because crop genome sequences are available, fine physical maps have become a popular alternative, mainly because of their faster development and almost unlimited resolution (i.e., at the base-pair level).

(c)    The identification of marker–trait associations. As mentioned, the genetic linkage between the trait of interest and the marker is a key aspect for marker-assisted breeding. The success in breeding programs can only be guaranteed if markers are tightly linked to the genes or QTLs, or closely associated with the target traits.

The most popular methodology to identify genes or genomic regions (QTLs) controlling target traits is referred to as "gene mapping" or "QTL mapping" (Figure 2). These mapping analyses are based on the identification of molecular markers that co-segregate with the trait of interest in an experimental population derived from a bi-parental cross. Different types of populations can be used, such as F2 populations, double-haploid (DH) populations, backcross, or recombinant inbred line (RIL) families. However, these populations present certain drawbacks; firstly, because the mapping resolution depends on the amount of recombination events happening during the population generation [28], which is usually limited; and secondly, because only those loci showing allelic variation between the parental lines can be analyzed. These limitations can be partially overcome by using multiparent RILs (as MAGIC populations: Multiparent Advanced Generation

InterCross), which increases both the allelic variability considered for analysis and the number of recombination events occurring during population development [29].

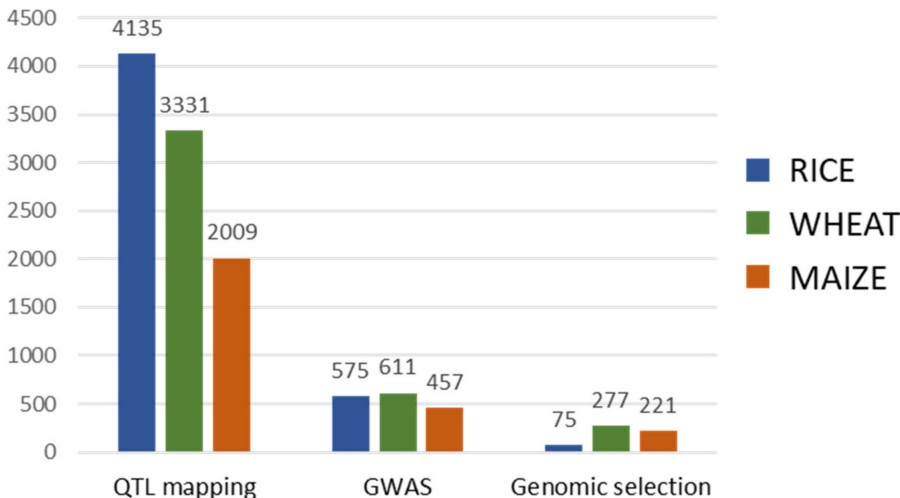

**Figure 2.** Number of bibliographic records referring to the main approaches for marker-assisted breeding in rice, wheat, and maize (source: Web of Science database, all years available; accessed 15 December 2020). QTL: Quantitative Trait Locus; GWAS: Genome-Wide Association Studies.

At the beginning of the 21st century, Genome-Wide Association Studies (GWAS) emerged as a powerful complementary mapping tool [30,31]. The GWAS approaches test the statistical association between the markers, usually SNPs, and the phenotype of a target trait scored in large panels of distantly related individuals (i.e., natural populations) or lines (i.e., wide germplasm collections including landraces or cultivar representatives). Compared to the family-based traditional mapping methods, GWAS not only hugely increases the allelic diversity covered by the analysis, but also improves the map resolution by taking into account all genetic recombinations that historically occurred in the generation of the (unrelated) genotypes composing the mapping panel [32]. Nevertheless, the genetic structure of the mapping population must be taken into account to avoid false, inaccurate associations [33]. An additional limitation of GWAS is that most of the association methods only consider bi-allelic variants and the underlying function of multi-allelic sites, which if not properly handled, may be ignored [34,35]. Efforts are being made in order to address this drawback [36]. It can be noted that the GWAS approaches have benefited from the parallel implementation of high-throughput sequencing technologies, such as Genotyping by Sequencing (GBS), that allow for the rapid and cost-effective genotyping of thousands of molecular markers [37].

Once a statistically significant association between a marker (or a group of markers) and a crop trait is demonstrated, it is assumed that the marker is located within or close to a genic sequence causative of the phenotype of interest. This is usually the first step to clone and functionally characterize genes. However, as already mentioned, even with limited or no knowledge of the biological function of the marked gene, the marker can be successfully applied for MAS.

Some economically important crop characteristics are controlled by one single gene or a few genes having a major effect on the plant phenotype. That is the case, for instance, of many resistances to pests/diseases, male sterility, self-incompatibility, and certain characteristic related to plant morphology or quality. For these, namely qualitative traits, the mapping approaches usually yield reliable markers and MAS designs are currently common in breeding programs. But many relevant traits, of quantitative inheritance, have a complex genetic architecture as being controlled by multiple genes or QTLs with small individual effects on the phenotypic variation. This holds for yield components as well as for most morphological and physiological plant characteristics determining crop responses

to abiotic limitations. Theoretically, all the QTLs contributing to the quantitative trait of interest could be taken into account in an MAS-based breeding scheme. However, the main obstacle remains to be the lack of efficient, reliable markers. For polygenic traits, non-genotypic factors usually have a great influence on the plant phenotype, with QTL × E and multiple epistatic interactions also being possible. This seriously hinders the validation of putative causative associations between markers or mapped QTLs and trait phenotypes [27].

When the association between genes/QTLs and traits has been established, marker-assisted backcrossing (MABC) is regarded as the simplest form of MAB. MABC facilitates the transfer of one or a few favorable genes or QTLs from one individual used as the genetic source (the donor parent, that may be agronomically unsuitable) into the elite breeding line or superior cultivar to be improved for the associated trait (the recurrent parent). Unlike traditional backcrossing, based on phenotypic presentation of the trait of interest, MABC is focused on the presence/absence of alleles of the markers linked with the target favorable gene(s)/QTL(s). MAS can also aid the concurrent selection of the elite parent alleles, representing a clearly time-saving strategy [38]. Marker-assisted gene pyramiding is another useful MAS application that allows the breeding of a novel cultivar by selecting gene/QTL-allele-linked markers of different donors mediating multiple-parent crossing or complex crossing, back-crossing, and recurrent selection. It may be a successful strategy to enhance a quantitatively inherited trait [27]. One of the most problematic issues in crossing breeding programs is linkage drag, which refers to the reduction in quality, yield, or other elite traits that can result from the introduction of deleterious genes or QTLs of a donor parent together with the favorable gene. It must be noted that MAS also offers a tool for monitoring undesirable genes in the offspring.

### 2.2. Genomic Selection

The sequencing of the whole genome and the development of powerful bioinformatic tools for associating characters and molecular markers has led to the design of new, more ambitious tools in order to assist in breeding programs.

Genomic selection (GS) or genome-wide selection (GWS) is a methodology based on MAS but characterized by the simultaneous selection for numerous markers (until hundreds of thousands) that cover the entire genome. Thanks to the high density of markers distributed along the genome, all genes/QTLs are expected to be in linkage disequilibrium with at least some of the markers [39]. GS programs consist of two distinct, consecutive, phases [40]. During the first phase, the effect on the phenotype of allelic variation at multiple loci is estimated and a "genomic estimated breeding value" (GEBV) formula is developed. For this, phenotypes and genome-wide genotypes of a reference population (called the training population) are analyzed, and significant associations between phenotypes and genotypes are prognosticated by means of statistical analyses. In the second phase, the selection of desirable individuals within a breeding population is carried out by firstly determining the genome-wide molecular marker profiles of candidate individuals, and then using the GEBV equation previously obtained. Crop breeding has already benefited from GS research (see Section 2.3. [40]), which is clearly more intense in wheat and maize than in rice, among the main crops (Figure 2). Some currently addressed issues (for example, the approximation of non-additive genetic effects and the combination of multiple traits or environments) will be key for improving the accuracy of GS predictions [41].

### 2.3. Marker-Assisted Breeding Oriented to Crop Improvement for Abiotic Limitations' Response

The timing of application of the main MAB approaches to drought tolerance (*sensu lato*, also including water-use efficiency and related topics), heat tolerance, and NUE in rice, wheat, and maize is showed in Table 1. For temporal contextualization, the year of publication of the first article where QTL mapping, GWAS and GS methodologies have been realized (not merely mentioned) on any breeding topic are also showed. It supports that water stress has usually been the first research goal among abiotic limitations but that,

despite their main relevance, the topics focused here have not always been among those initially addressed. Among crops, maize improvement received an earlier interest for the application of QTL mapping and GS approaches, despite the fact that the overall research efforts have been lesser than in wheat according to bibliometrics (Figure 2).

**Table 1.** Year and bibliographic reference of the first documented application of different marker-assisted breeding approaches in rice, wheat, and maize.

| Approach | Topic | Rice | Wheat | Maize |
|---|---|---|---|---|
| QTL mapping | (Any) | 1990 [42] | 1992 [43] | 1987 [44] |
| | Drought tolerance | 1995 [45] | 1994 [46] | 1995 [47] |
| | Heat tolerance | 2000 [48] | 2002 [49] | 1991 [50] |
| | NUE | 2001 [51] | 2004 [52] | 1999 [53] |
| GWAS | (Any) | 2009 [54] | 2009 [55] | 2011 [56] |
| | Drought tolerance | 2010 [57] | 2014 [58] | 2011 [59] |
| | Heat tolerance | 2017 [60] | 2017 [61] | 2019 [62] |
| | NUE | 2019 [63] | 2014 [64] | 2017 [65] |
| Genome selection | (Any) | 2014 [66] | 2011 [67] | 2007 [68] |
| | Drought tolerance | 2018 [69] | 2018 [70] | 2013 [71] |
| | Heat tolerance | - | 2018 [70] | 2019 [72] |
| | NUE | 2016 [73] | 2019 [74] | 2015 [75] |

Some features of QTL mapping, GWAS and GS in each of these crops and technical achievements regarding sustainability-related traits are described in the following sub-sections.

2.3.1. Rice

Rice is the most drought-sensitive cereal. Thus, improvement of water-use efficiency and tolerance to water deficits are the main rice breeding goals. Two of the earliest reports on QTL mapping in rice did indeed deal with the identification of genome regions involved in traits related to plant responses to drought, i.e., root morphology and osmotic adjustment [76]. The long list of crop sustainability related traits for which QTLs or genes have been mapped in rice is headed, firstly, by traits improving plant response to drought, and secondly, by heat tolerance and traits related to nitrogen-use efficiency NUE (Figure 3) [63,77–80]. However, it also includes morphological or physiological characteristics related to salinity tolerance [81–83]; photosynthetic efficiency [84,85]; phosphorous and potassium use efficiency [86,87]; tolerance to deficiency in other essential nutrients [88]; or plant growth under aluminium stress [89], among others. The contemporary application of molecular markers to accelerate breeding programs is not a potential advantage of promising modern approaches, but a current selection strategy in rice, also for sustainability-related traits [90–92].

GWAS is relatively underused for the genetic dissection of rice traits compared to the predominance of this crop over wheat and maize, regarding the application of QTL mapping (Figure 2) and most of the modern analytical tools (see below). This is particularly remarkable considering that association studies are facilitated when dense genetic or physical maps are available in the species, and that, as noted above, the rice genome was well assembled years in advance of the maize and the wheat genomes. A possible explanation can be related to the easier construction of genetic maps in diploids, compared to polyploid species [93]. It may make GWAS comparatively less advantageous than other mapping tools in rice than in crops with more complex genomes such as maize (an ancient polyploid) and, specially, wheat. Nevertheless, several studies have taken advantage of its applicability for the easy identification of novel QTLs involved in plant response to environmental constraints by exploring wide rice germplasm collections of diverse geographical origins [73,94]. As noted for GWAS, genome selection will likely have greater impacts as breeding assisting tools in crops with more complex genomes (Figures 2 and 3),

but its predicting ability for developing rice varieties adapted to dryer environments has also been demonstrated [69,95].

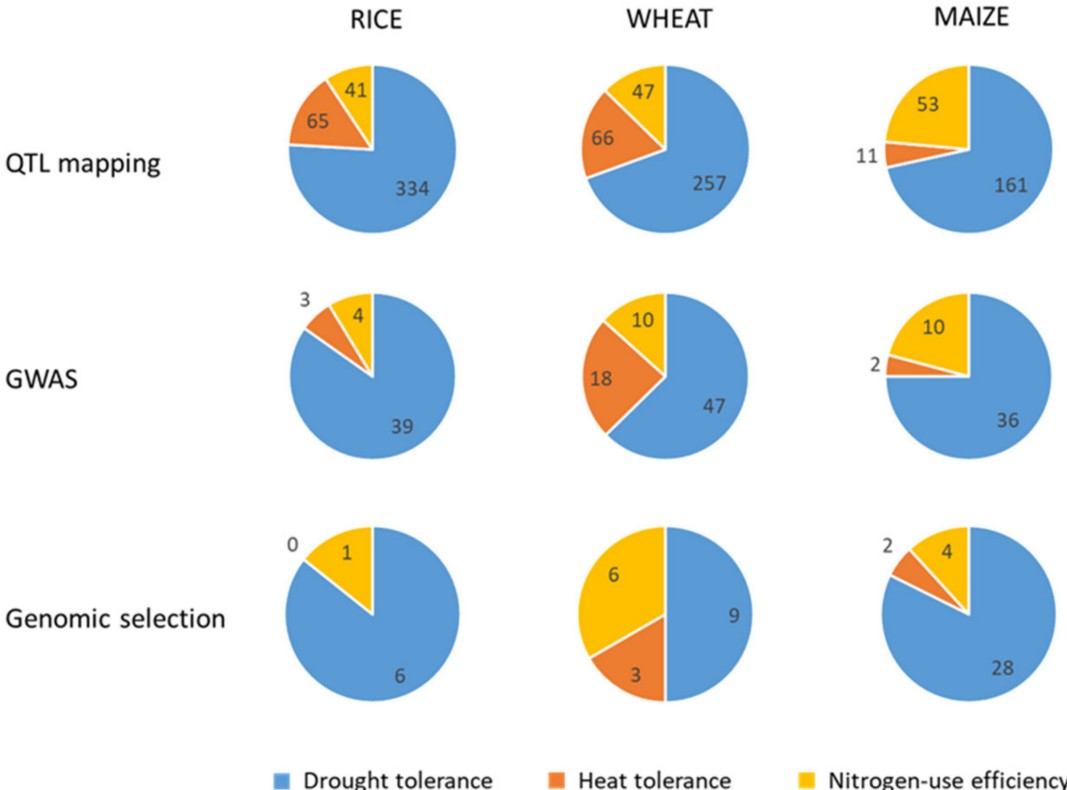

**Figure 3.** Relative relevance of breeding topics related to the main abiotic stresses and nutrient limitations in bibliographic records referring to QTL mapping, GWAS, and genomic selection, in rice, wheat, and maize (source: Web of Science database, all years available; accessed on 15 December 2020).

### 2.3.2. Wheat

The first studies to determine the genome location of QTLs in wheat analyzed the association between the phenotype for the quantitative trait of interest and morphological or isozyme traits (i.e., polymorphisms for qualitatively-inherited genes) that had been already mapped by using the Sears' cytogenetic stocks [43] (Table 2). These and other cytogenetic stocks derived of cv. Chinese Spring were also key materials to map the molecular markers (i.e., polymorphisms for DNA sequences) that were subsequently employed for QTL mapping by means of genetic linkage analyses [46,96]. In a comprehensive review, Gupta and coworkers have compiled the many QTLs related to agronomic, physiological, and root-related traits that have been identified under water, heat, and salinity stresses in wheat identified to date [97]. The mapping of wheat genome regions related to nutrient-use efficiency has received less attention (Figure 3). Nevertheless, QTLs with a significant effect, not only on biomass, yield or grain protein content, but also on N uptake and N utilization efficiency, have been detected under different nitrogen regimes [98,99].

Compared to rice and maize, the emergence of GWAS is relevant as the alternative to the classical biparental-population mapping (Figure 2), especially for the location of drought and heat tolerance-related QTLs in the complex wheat genome (Figure 3) [100]. The much greater variability detected at the genomic level in local landraces compared to commercial wheat cultivars makes this approach the most useful for the identification of novel alleles involved in abiotic stress tolerance and adaptation to low-input management practices [101]. Once an accurate and complete wheat genome sequence has been released, the availability of increasingly more reliable high-density SNP arrays [102] will prompt mass genotyping for GWAS approaches.

Wheat is likely the crop for which more studies confirming the applicability of molecular markers to assist selection programs are available. However, most of the documented examples refer to markers linked to breadmaking quality traits and pathogen resistances [103,104]. Breeding for complex polygenic traits such as tolerance to water stress or other abiotic constrains is still conducted by the direct selection of better performing plants in limiting conditions [105]. Nevertheless, the panel of new tools implemented after the wheat genome has been fully sequenced has boosted the design of predictive models and novel genome selection-based breeding strategies (Figures 2 and 3), some of which are being tested in several international programs. A worldwide leading institution, CIMMYT (International Maize and Wheat Improvement Center), is following a genome selection strategy for wheat quality breeding [106,107]. Promising results have already been obtained in the scope of durum wheat adaptations to drought environments by ICARDA (International Center for Agricultural Research in the Dry Areas) [108] as well as for wheat N-use efficiency improvement [74].

### 2.3.3. Maize

Maize was the earliest main crop where QTL mapping studies were conducted [44] (Table 1). Since then, genetic loci associated to traits involved in plant response under environmental constraints such as drought, heat, salinity, and waterlogging, or under nitrogen deficit, have been identified by using either biparental populations or panels of maize lines encompassing wider genotypic variability [53,62,109–112]. The bibliometric analysis evidences a higher relevance of NUE and related traits in QTL mapping studies conducted in maize compared with rice and wheat (Figure 3).

Despite a lower abundance of research documents, maize has been pioneering in the breeding-oriented application of molecular markers, which is surely related to the higher profitability of biotech investments for hybrid cultivar improvement. So, as early as in 1992, the usefulness of marker-assisted selection of agronomic and yield-related quantitative traits was demonstrated in maize [113]. For context, the early uses of MAS in rice and wheat were not only later, but associated to monogenic traits or based on markers linked to alleles of known major effects on the traits of interest [114,115]. Maize was also the first, among these main crops, where experimental data were used to test the accuracy of breeding-by-design models [68] (Table 1). Since then, the ability of a number of genomic prediction indexes to select superior maize lines in varied stressed environments has been documented [71,72,75].

## 3. Gene Expression Analysis

### 3.1. Gene Expression Analysis Approaches

For decades, gene expression studies were almost exclusively based on the identification of individual proteins by electrophoretic methods or enzymatic activity assays [116–118]. However, over the past 20 years, the field of gene expression profiling has undergone a dramatic revolution with the incorporation of analytical tools that can provide a dynamic picture of the complete set of transcripts in specific tissues of an organism. The transcriptome includes the messenger RNA (mRNA) and non-coding RNA (ncRNA) molecules. Unlike the genome, which is nearly static for each organism, the transcriptome is active and dynamic, and can change in a specific cell, tissue, or organ according to the developmental stage or in response to external stimuli. Thus, transcriptomics is considered to be the major large-scale platform for studying the biological functioning of a living organism. By analyzing the transcriptome, researchers can determine which sets of genes are turned on or off in a particular condition and can quantify the changes in gene expression among different biological contexts [119].

Numerous molecular biology techniques have been used for transcription quantification and expression profiling [120]. The first of these techniques (Northern blotting, in situ hybridization and Reverse transcription polymerase chain reaction (RT-PCR)) served to determine the presence of single or few transcripts in a qualitative way [121,122]. A

posterior technical improvement that allows the detection of mRNA present even at low levels (real-time RT-PCR or Quantitative PCR (qPCR)) has been the most widely used for absolute and relative gene expression quantification [123,124]. The expression of thousands of known genes can be studied simultaneously since the development of microarrays, which consist of a collection of specific-DNA spots attached to a solid surface that are hybridized with a cDNA or cRNA sample [125]. This methodology has provided a powerful tool for monitoring global changes of gene activity, needed to understand key molecular pathways involved in plant development and in physiological responses to external constraints [126,127]. More recently, NGS-based RNA sequencing (RNA-seq) has entailed another revolution in gene expression analysis [128]. RNA-seq uses NGS to determine the presence and quantity of RNA in a biological sample at a given moment, allowing a very sensitive monitoring of the changes in the cellular transcriptome during development processes or under biotic or abiotic stresses. In addition to mRNA transcripts, RNA-seq can analyze different populations of RNA including lncRNA (long ncRNA) and small RNA, such as micro-RNA, transfer RNA, and ribosomal RNA. The advent of high-throughput RNA sequencing technologies has also made it possible to map transcripts onto the genome for studying the structure of genes and allele variants, alternative splicing patterns, or post-transcriptional modifications [4,128].

All these molecular techniques have been extensively used for gene expression profiling in crop plants, resulting in the identification of key genes and pathways regulating traits of breeding interest. Particularly remarkable is their ability to identify genes coding for transcription factors (TFs), regulatory proteins that control the turning on/off of multiple downstream genes which are coordinately regulated by external or developmental signals [129]. TF genes are assumed to be the best breeding target for the successful improvement of polygenic crop traits [130].

**Table 2.** Year and bibliographic reference of the first documented application of RNA-based approaches in rice, wheat, and maize. The timing of gene expression analyses by crop and topic is also indicated.

| Approach | Topic | Rice | Wheat | Maize |
|---|---|---|---|---|
| Gene expression | (Any) | 1984 [118] | 1972 [117] | 1971 [116] |
|  | Drought tolerance | 1988 [131] | 1991 [132] | 1991 [133] |
|  | Heat tolerance | 1991 [134] | 1992 [135] | 1993 [136] |
|  | NUE | 2006 [137] | 2008 [138] | 2006 [139] |
| Transcriptomics | (Any) | 2001 [140] | 2002 [141] | 2003 [142] |
|  | Drought tolerance | 2006 [143] | 2009 [144] | 2007 [145] |
|  | Heat tolerance | 2005 [146] | 2007 [147] | 2015 [148] |
|  | NUE | 2006 [137] | 2008 [149] | 2009 [150] |
| Quantitative PCR | (Any) | 2003 [151] | 2003 [152] | 1999 [153] |
|  | Drought tolerance | 2008 [154] | 2009 [155] | 2007 [156] |
|  | Heat tolerance | 2008 [154] | 2011 [157] | 2007 [156] |
|  | NUE | 2007 [158] | 2013 [159] | 2011 [160] |
| RNA-seq | (Any) | 2010 [161] | 2011 [162] | 2011 [163] |
|  | Drought tolerance | 2015 [164] | 2014 [165] | 2012 [166] |
|  | Heat tolerance | 2015 [167] | 2015 [168] | 2015 [169] |
|  | NUE | 2018 [170] | 2014 [171] | 2015 [172] |

Figure 4 shows the number of research documents referring to transcriptomics and the main modern methodologies for gene expression analysis (i.e., qPCR and RNA-seq) in rice, wheat, and maize. Like for most other molecular breeding tools, rice is the crop where these approaches have been more extensively used, but the distance with wheat and maize has narrowed considerably regarding the most recent technical incorporations.

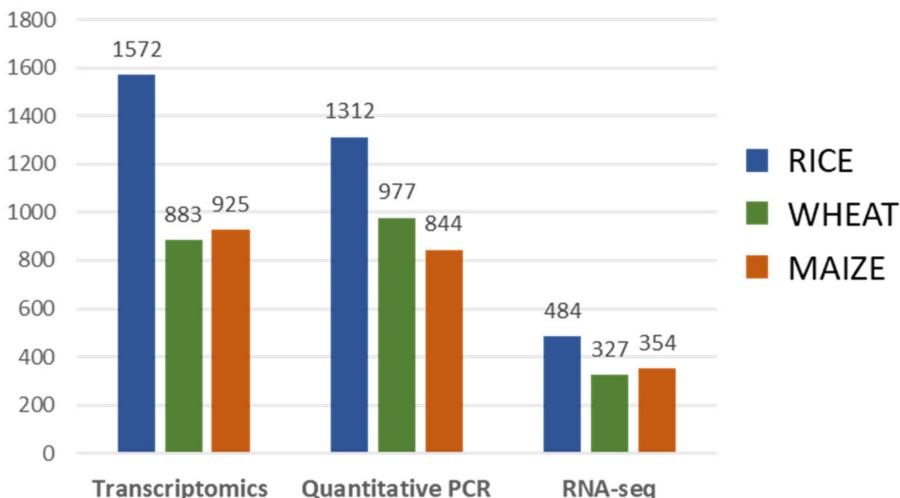

**Figure 4.** Number of bibliographic records referring to main modern approaches for gene expression analysis in rice, wheat, and maize (source: Web of Science database, all years available; accessed on 15 December 2020).

### 3.2. Gene Expression Analysis Oriented to Crop Improvement for Abiotic Limitations' Response

Table 2 shows the timing of application of molecular approaches based on mRNA expression to the functional analysis of genes or QTLs putatively involved in drought tolerance, heat tolerance and NUE in rice, wheat, and maize. For temporal contextualization, the year of publication of the first articles which documented gene expression analyses are showed, in addition to the year of publication of the first reports describing a real use of these novel methodologies to any breeding topic.

It is worth noting the lack of NUE-related gene expression studies in rice and wheat prior to the implementation of transcriptomic approaches, although no other temporal biases can be clearly established either for focused topics or crops. Some usage particularities and technical achievements regarding sustainability-related traits in rice, wheat and maize are described in the following.

#### 3.2.1. Rice

As already noted, global expression studies using transcriptomic approaches, combined with bioinformatics, can serve to identify novel genes putatively involved in favorable responses. An early version of this strategy was applied to the genetic dissection of drought tolerance in rice as early as in 1988 [131], more than 10 years before the term "transcriptomics" was first used in plant studies [173] (Table 2). Those "pre-omic"-era studies were based on mRNA in vitro translation or protein extraction and sequencing for further comparison with the existing protein databases available. Interestingly, the conceptually equivalent evolved approach, RNA-seq, has demonstrated the significant effect of water-by-nitrogen interactions at the gene expression level [174]. It is also worth noting that several among the first rice studies where cDNA microarrays were employed to demonstrate differences in transcriptional profiles between genotypes or treatments allowed the identification of genes that were up- or down-regulated in abiotic stress conditions, different from those reflected in Table 2 such as salinity and nutritional starvation [175,176]. The use of qPCR for quantifying the differential expression of specific rice genes involved in plant response to major abiotic constrains is also documented (Figure 5 and Table 2). This is another example of a molecular analytical tool whose first report in rice was oriented to the study the functional genetic basis underlying the plant response to an abiotic limitation, i.e., phosphorous starvation [151].

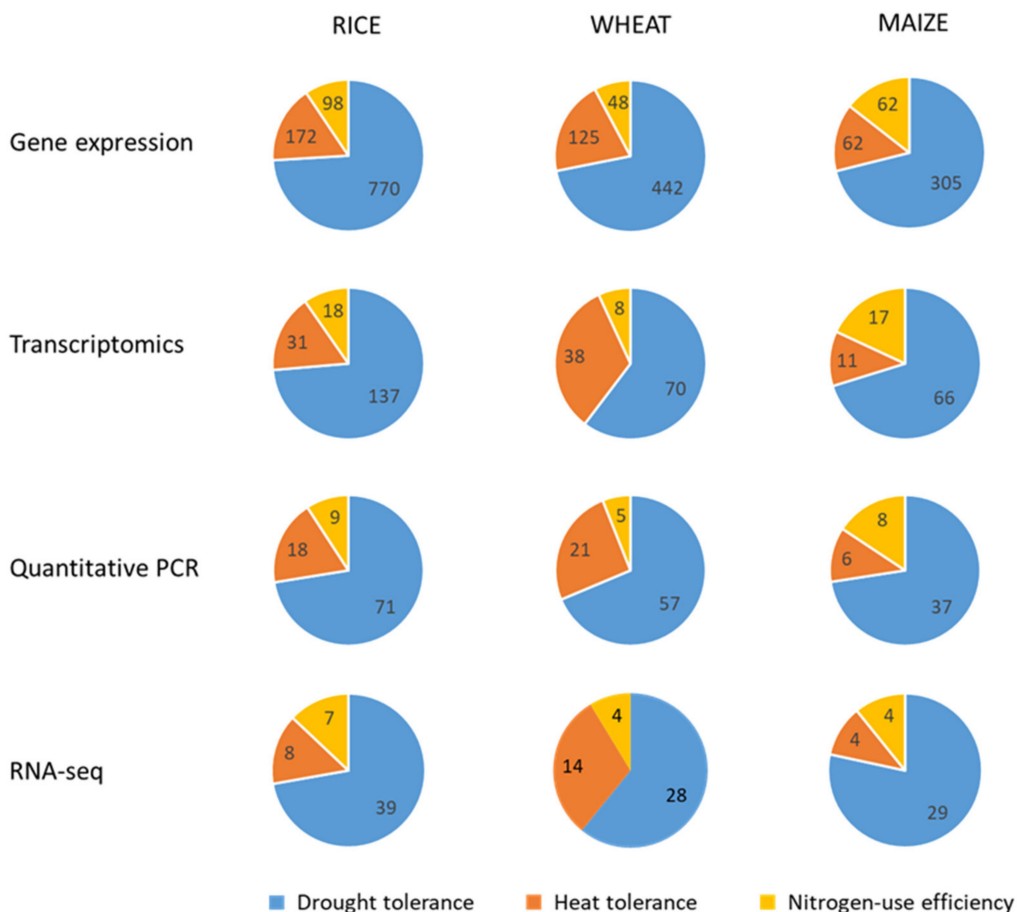

**Figure 5.** Relative relevance of breeding topics related to major abiotic stresses and nutrient-limiting topics in bibliographic records referring to gene expression and related tools in rice, wheat, and maize (source: Web of Science database, all years available; accessed on 15 December 2020).

### 3.2.2. Wheat

In wheat, the differential expression of genes depending on external conditions was initially studied only for Mendelian-inherited (monogenic) traits, and deduced from the associate phenotypic change [117]. By the early 1980s, the up-/down-regulation of specific wheat genes began to be determined by in vitro translation or cloning, as cDNA, of messenger RNAs extracted from treated cells [177]. Dozens of studies have used transcriptomic approaches to demonstrate the active role of TFs and genes involved in cellular defense mechanisms, mainly against oxidative or osmotic stress, in the wheat plant response to abiotic limiting conditions [178–181]. The transcriptomic studies dealing with the functional characterization of wheat genes presumably involved in the response to low N input are significantly scarce (Figure 5). However, the differential tissue-specific expression of several genes related to N assimilation, N and C metabolism, or coding transcription factors, has been demonstrated and quantified in durum wheat cultivar Svevo [182].

### 3.2.3. Maize

The first documented reports of changes in mRNA synthesis were conducted before the transcriptomic era and corresponded to plastid but not nuclear maize genes [183,184]. Since then, differential gene expression analyses at the transcriptome level have helped to identify many maize genes that are up- or down-regulated by environmental stresses [145,148,185], or by nitrogen levels [186,187]. Studies combining QTL mapping with the characterization of transcripts that are differentially expressed under abiotic stressed conditions have been especially successful in identifying gene targets for the breeding of maize drought-tolerant

cultivars [188,189]. The role of genes involved in C and N metabolism in plant drought response has also been evidenced by transcriptomic approaches [190]. Among the reports addressing the genetic basis of NUE in maize, it is worth noting the study of Yang et al., who determined that around 7% of the maize transcriptome is nitrogen-responsive [186]. These authors have developed a set of gene biomarkers whose expression profiles can be a cost-effective tool to monitor the nitrogen status of maize plants growing under field conditions. On the other hand, and as noted for rice, transcriptomic analyses have demonstrated that simultaneous water and nitrogen stresses have a greater impact on gene function than those expected according to the individual effects of these input limitations [191]. As for QTL mapping, it is again remarkable the relative relevance of gene expression studies addressing NUE and related traits in maize compared to rice and wheat, crops where thermotolerance occupies a clear second position in the research interest after drought tolerance (Figure 5).

## 4. Genetic Modification

The third main group of breeding assistant tools includes a number of techniques allowing that crop improvement can be based on genes or alleles that do not naturally exist within the crop species or within its sexually compatible relatives. These tools are supplementary to plant breeding and always need to be coupled with classical breeding program for evolving varieties of commercial value. It must be noted that some of them (i.e., transgenesis and gene editing) are not only employed to increase genetic variability but for gene function analyses that characterize how a specific change in the coding or regulatory elements of a gene alters the plant phenotype.

### 4.1. Mutagenesis

Mutagenesis cannot be strictly considered as a modern crop breeding tool. Chemical or radiation treatment of plants has been used since the second half of the 20th century to generate mutations that could give rise to interesting new alleles [192,193]. In fact, these techniques have been highly successful, with more than 3200 varieties of around 200 species having been obtained by mutagenesis over the last 70 years (IAEA/FAO database, https://mvd.iaea.org/, (accessed on 30 December 2020)). The type of mutations introduced by these techniques does not essentially differ from spontaneous mutations. A high percentage of induced mutations are simple nucleotide changes (single nucleotide polymorphisms, SNPs), although more extensive gene or chromosomal rearrangements may also occur [193]. All of these modifications occur essentially randomly in the genome, and the task of genetic breeding is to identify those responsible for novel variability of interest and eventually transfer them to yielding varieties. With this in mind, numerous mutant collections have been built in different crop species. A genomic approach, termed TILLING (Targeting Induced Local Lesions IN Genomes), that combines chemical mutagenesis with a sensitive mutation detection PCR-based strategy, has been the most successful to develop and to screen mutant populations for allelic variants in target genes [193–195].

### 4.2. Transgenesis

Transgenic plants are plants into which one or more genes (transgenes) have been artificially inserted using genetic engineering techniques. Usually, the entire transgene or some of its components (i.e., promoter, coding region) come from an unrelated plant or from an organism belonging to a completely different kingdom, including bacteria and viruses. However, in the approach referred as "cisgenesis" a gene from the same or a close species is first isolated and engineered, and then introduced back into the recipient organism [196,197]. Many distinct protocols have been developed since the early experiments conducted in the 1980s, but most genetically modified plants are obtained by either of two transformation methods: the biolistic approach (particle gun method) or mediated by *Agrobacterium tumefaciens* [198].

The main breeding-applied purpose of inserting transgenes in a crop plant is to introduce a trait which does not occur naturally in the species, in order to improve its agronomic performance or product quality. Since 1996, the process has materialized in more than 500 authorized genetically modified (GM) events, some of which are currently cultivated in a global area of around 200 million hectares [199]. Most GM cultivars correspond to maize, soybean, cotton and canola, and express herbicide tolerance and/or insect resistance; however, other crops and traits successfully engineered have also reached the market (see updates in the database of the International Service for the Acquisition of Agri-biotech Applications, ISAAA, at www.isaaa.org (accessed on 22 December 2020)).

However, transgenesis may also be used to test the presumed function of a crop gene by genetically manipulating either its coding region or the non-coding sequences that control the level and pattern, either spatial or temporal, of its expression (i.e., promoters, enhancers). These kinds of experiments, frequently conducted in model plants such as Arabidospsis or tobacco, have been essential to confirm the real involvement of mapped genes on crop traits [200,201].

Gene silencing and gene overexpression are the main transgenic strategies used to determine plant gene function. Both approaches aim to deduce the role of a gene by examining the phenotypic changes that are produced when its expression is altered. Gene silencing constructs inactivate the target gene by a phenomenon called RNA interference, by which the expression of antisense RNA (i.e., complementary to the RNAm sequence) or double-stranded RNA stretches triggers the enzymatic degradation of the target mRNA, that cannot be further translated [202]. Gene overexpression, usually achieved by inserting strong promoters upstream of the gene coding region, may be specially interesting when the target gene is functionally redundant with another gene (i.e., when the role of one of them may cover up the function of the other, and vice versa, in gene silencing mutants), or when the silencing of the gene is particularly deleterious or even lethal [203]. In a breeding-applied context, gene silencing has been very successful to improve relevant crop traits, such as increased shelf-life of fruits and flowers, enhanced nutritional quality of oil, or decreased lignin content of cellulosic biomass, among others; meanwhile, overexpression has been used to confer to plant resistance to herbicides or to different biotic stresses.

Figure 6 depicts a remarkable difference between the research documents on transgenic approaches in wheat and maize, which does not hold for any other of the tools analyzed. The likely reasons for this peculiarity are considered below.

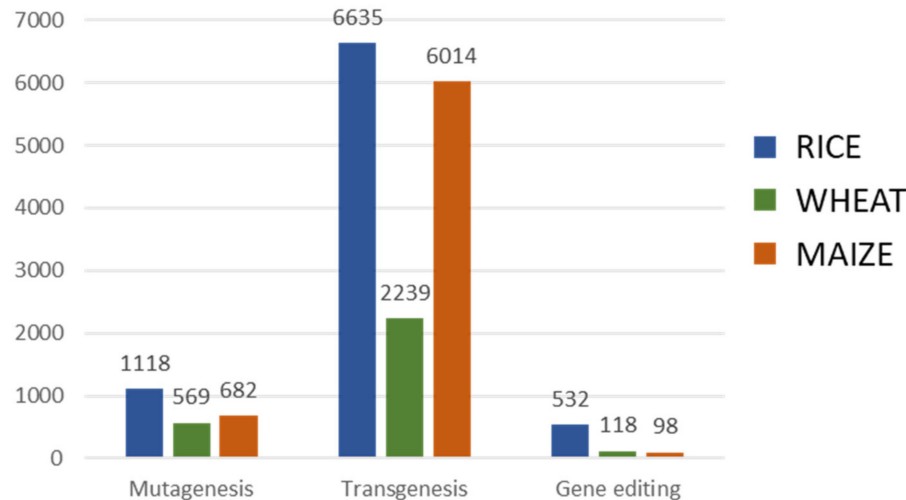

**Figure 6.** Number of bibliographic records referring to main approaches for genetic modification in rice, wheat and maize (Source: Web of Science database, all years available; accessed on 15 December 2020).

### 4.3. Gene Editing

Genome editing technologies address one of the most demanding challenges of current plant biotechnology: the custom design of new crop varieties. By using this set of tools, the repression or activation of gene expression, modifications of gene function, or the creation of gene knockouts can be performed, mediating the targeted manipulation of DNA sequences [204].

Gene editing requires that a nuclease enzyme introduces a double-stranded DNA break (DSB) at the site where a given gene wants to be edited. Afterwards, the end-joining or recombination pathways can repair the induced DSBs introducing simultaneously the desired modifications at the target locus [205]. However, off-target cleavage events and/or erroneous restoring of the physical integrity of the chromosomal DNA are far from negligible, and many target editing attempts provoke undesired mutations, even deleterious, or gross chromosomal rearrangements [206]. As in the other genetic modification techniques, screening procedures must be followed subsequently to identify plants that have been edited at the intended locus.

Until the generalization of editing systems based on the clustered, regularly interspaced, short palindromic repeats (CRISPR)-associated nuclease 9 (Cas9) [207], different classes of endonucleases have been used to design genomic editing tools, such as the mega-nucleases, the zinc finger nucleases (ZFNs), and the transcription activator-like effector nucleases (TALENs) [208–210]. Basically, these earlier systems consisted of chimeric nucleases with two functional domains: one of the domains, through binding to a specific DNA sequence, provides the specific action site to the nuclease domain, which catalyzes the cleavage of DNA [211]. TALEN-based tools have been the most used among these former gene editing systems. The main reason for their quick and full replacement by CRISPR/Cas-based tools is that TALEN site-specificity is based on the nuclease sequence, and a new protein must be designed for targeting each gene or gene location, while CRISPR/Cas specificity is based on the sequence of a guide RNA, which makes the system simpler and more versatile at a lower cost [204]. It is likely that the limited usage of any these approaches in wheat and maize (Figure 6) can be due to the easier molecular dissection of genes and their functions in rice, which is a pre-requisite for a suitable design of gene manipulation strategies.

### 4.4. Genetic Modification Oriented to Crop Improvement for Abiotic Limitations' Response

The timing of application of mutagenesis, transgenesis and gene editing tools in rice, wheat, and maize, oriented to the improvement of drought tolerance, heat tolerance and NUE, is showed in Table 3. Again, for temporal contextualization, the year of publication of the first article where these genetic modification methodologies have been realized on any breeding topic are also showed.

**Table 3.** Year and bibliographic reference of the first documented use of mutagenesis, transgenesis and gene editing tools in rice, wheat, and maize.

| Approach | Topic | Rice | Wheat | Maize |
|---|---|---|---|---|
| Mutagenesis | (Any) | 1971 [212] | 1964 [213] | 1961 [214] |
| | Drought tolerance, WUE | 2007 [215] | 2001 [216] | 2014 [217] |
| | Heat tolerance | 2003 [218] | 2014 [219] | 2020 [220] |
| | NUE | 2011 [221] | - | 2006 [139] |
| Transgenesis | (Any) | 1988 [222] | 1990 [223] | 1988 [224] |
| | Drought tolerance, WUE | 1998 [225] | 2000 [226] | 2002 [227] |
| | Heat tolerance | 2000 [228] | 2008 [229] | 2007 [156] |
| | NUE | 1997 [230] | 2001 [231] | 2015 [232] |
| Gene editing | (Any) | 2012 [233] | 2013 [234] | 2014 [235] |
| | Drought tolerance, WUE | 2017 [236] | - | 2017 [237] |
| | Heat tolerance | 2020 [238] | - | - |
| | NUE | 2018 [239] | - | 2020 [129] |

According to the bibliographic and bibliometric analyses performed, and despite its relatively early introduction as breeding-assistant tool in the major crops (Table 3), the use of mutagenesis for the specific breeding goals of interest in the present study has been really limited, with a total of 36, 22, and 8 documents on rice, wheat, and maize, respectively, regarding drought, heat tolerance, or NUE. For that reason, this approach has not been further considered. The time gap between the first use of transgenesis in the major crops and its first application to abiotic limitations, of around 10 years, is the longest among all the molecular breeding approaches considered (Tables 1–3). This can be attributed to the mostly polygenic nature of plant responses to environmental conditions. It greatly hinders the structural and functional characterization of single genes with a significant effect on the target trait, always required for implementing genetic modification strategies. Some technical particularities and achievements regarding each of the crops under focus are described in the following sub-sections.

### 4.4.1. Rice

Most of the earliest studies that followed transgenic approaches for the functional characterization of crop genes actually engineered model plants with constructs carrying coding regions or control elements of the gene of interest. In the case of rice, one of these pioneering applications of transgenesis used transgenic tobacco plants to study the function of a glutamine synthetase rice promoter [200]. Since then, many rice genes putatively conferring tolerance to drought and other limiting plant stresses have been functionally characterized either in heterologous systems (i.e., tobacco, Arabidopsis) and in transgenic rice [240] (Figure 7). Rice, as a model monocot for transformation experiments, has also been engineered to demonstrate the protecting function of alien crop genes against abiotic stresses such as drought and salinity [241,242]. Gene editing has been the latest methodology incorporated to the toolbox for the functional characterization of genes. Its use for plant traits in the scope of this review, is comparatively novel (Table 3) and still scarce (Figure 7); however, a modified CRISPR protocol has already successfully inactivated a rice gene regulating the stomatal density, an important determinant of water use efficiency [243].

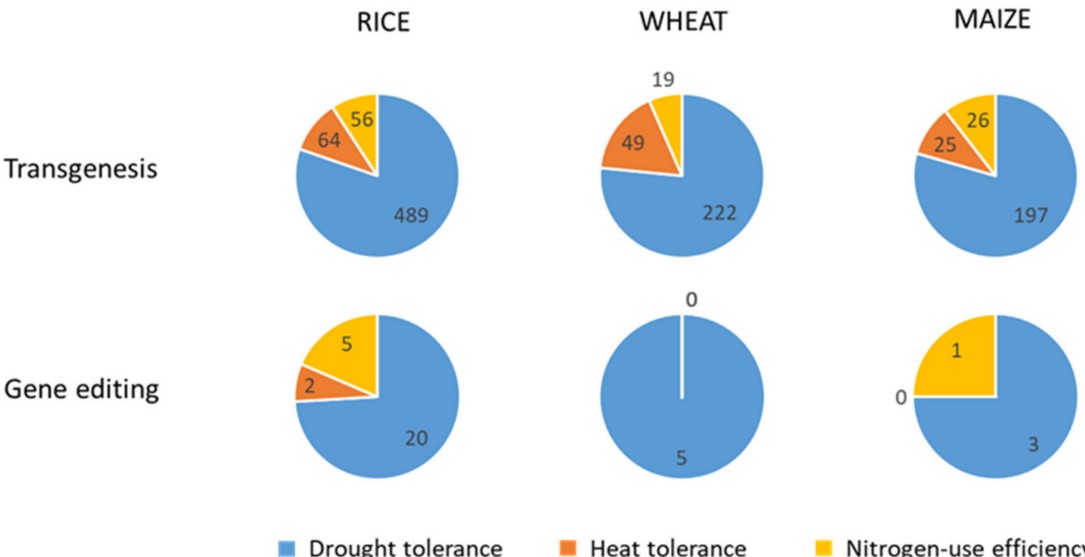

**Figure 7.** Relative relevance of breeding topics related to main abiotic stress and nutrient-limiting topics in bibliographic records referring to transgenesis and gene editing tools in rice, wheat, and maize (source: Web of Science database, all years available; accessed on 15 December 2020).

In addition to the genetic engineering experiments strictly oriented to basic-research purposes, transgenic rice plants have also been created with the applied goal of developing

genetically modified (GM) commercial varieties. However, the current number of rice GM varieties is very limited according to the information in the most complete, cumulative, database of the International Service for the Acquisition of Agri-biotech Applications (ISAAA, www.isaaa.org, accessed on 22 December 2020). There are only eight rice events out of the 526 events ever approved in any country; and all, except the famous "Golden Rice", carry transgenes related to biotic stresses, i.e., insect resistance and herbicide tolerance. A major reason for the limited representation of rice, compared to maize, among GM varieties will be considered below, but the null presence of abiotic stresses among the target traits is very likely due to the already noted complex genetic basis of the plant physiological response in limiting abiotic conditions. This makes it unlikely that single genes which may have demonstrated significant tolerant effects when checked under controlled experimental conditions can actually improve crop performance in a real field environment [244].

### 4.4.2. Wheat

Due to the technical problems to transform monocots, the first uses of transgenesis to analyze the function of wheat genes were made by transforming the model dicot tobacco with wheat genes and analyzing the expression of different constructs [245]. A 1989 report transformed rice instead of tobacco, for the first time, to identify functional elements of a wheat gene regulated by the phytohormone abscisic acid [246]. After several transforming protocol attempts [223], it was by 1992 when the first transgenic fertile wheat plants were reported, the last among the major crops [247]. In wheat, the transgenic approaches have evidenced a special interest on sustainability-related traits, drought tolerance and heat tolerance, representing 10% and 2% of the total documents, respectively; meanwhile, the corresponding values are 7.4% and 1% in rice, and 3.3% and 0,4% in maize (see data in Figures 6 and 7). Most genes have been engineered with the final aim of developing abiotic stress-tolerant wheat cultivars code transcription factors, i.e., regulatory proteins that control and coordinate the expression of multiple genes in response to external stimuli [248,249]. This is the more promising strategy due to the complex genetic basis governing the different physiological pathways that underlie crop performance in real field conditions that, in the case of wheat, is additionally hindered by the existence of three homoeologous copies of many functional genes. Regarding the improvement of N use efficiency, the successful introduction into a winter wheat variety of a favorable allele of the wheat GS2 gene, coding the plastidic isoform of the glutamine synthetase enzyme has been reported [250]. Whether any of these developments will materialize in the next GM wheat cultivars cannot be predicted, but the history of GM wheat approvals is even more unsuccessful than that of rice, with only a single herbicide-tolerant event in the GM database of the ISAAA (https://www.isaaa.org/gmapprovaldatabase/; accessed on 22 December 2020). TALEN and CRISPR systems have been used for gene editing of a number of wheat genes since a pioneering work on cell suspensions conducted in 2013 [234]. However, the induction of targeted mutations on genes involved in the traits under focus is not yet documented in wheat (Table 3), despite a few reports having claimed the suitability of this approach for addressing the creation of drought-tolerant cultivars (Figure 7).

### 4.4.3. Maize

As described for rice and wheat, the first uses of transgenesis for functional analyses of maize genes were made by transforming dicots (petunia and tobacco) with maize genes and analyzing the expression of different genic constructs. Among these early reports, attempts were made to dissect the molecular basis of plant response to an abiotic constraint such as high temperature [251]. Currently, maize is by far the crop most represented among the GM cultivars, with 283 authorized events in the GM database of the ISAAA (https://www.isaaa.org/gmapprovaldatabase/; accessed on 22 December 2020), which may explain the particular abundance of transgenic research in maize (Figure 6). The greater interest of breeding companies and institutions to develop GM-maize but not GM-rice or GM-wheat varieties can be ultimately attributed to the already mentioned predominance of

hybrid elite cultivars in maize. Herbicide tolerance and insect resistance are the commercial traits more frequent in maize GM cultivars, with almost 90% of authorized events carrying transgenes that protect plants against one or both of these biotic stresses. A number of engineered maize lines have been reported to perform with enhanced tolerance to drought, heat, salinity, or NUE [227,232,252–255], since the first GM maize patent related to abiotic limitations was issued in 2000 [254]. However, the interest in these topics seems relatively low (comparing data for maize in Figures 6 and 7) and, to date, there is a single genetic engineered construction that has been successful to develop GM maize commercial cultivars with improved abiotic stress tolerance. That construction carries a bacterial gene coding for an RNA chaperone protein whose expression has been demonstrated to maintain cellular functions under water stress conditions [256]. This transgene is currently present in seven authorized GM events, most of which carry additional constructions for herbicide tolerance and/or insect resistance. Successful attempts based on targeted mutagenesis approaches (TALENS or CRISPR systems) have also been reported either for drought tolerance or NUE [129,257] (Table 3), but are still limited compared to the tool usage in rice (Figure 7).

## 5. Future Prospects

The improvement of crop plant response to the main environment constraints and nutrient limitations represents a major breeding goal to assure food security in the future, particularly under the threats of the changing climate and the expanding population [258]. The low success in creating abiotic stress-tolerant cultivars, despite the great number of genes for which a protective function has been demonstrated in controlled conditions, has been considered in earlier reviews [244,259]. Particularly, the actual reliability of phenotype data if real field conditions are replaced by high-throughput phenotyping platforms has been pointed out as the bottleneck for a confident application of the many localized markers [260]. Wide germplasm collections, including accessions adapted to unfavorable environments or low-input cultivation management (i.e., landraces, and local varieties), are likely the haystack where the needles will be found [15,101,261]. The current ability to cheaply sequence thousands of materials together with the growing development of bioinformatics tools will surely provide breeders with finer tuning of GS strategies [262], acknowledged as the most promising approach for crop improvement for traits that involve complex plant physiological responses [40]. As far as genetic modification is concerned, the current list of engineered sustainability-related traits in commercial varieties is almost limited to herbicide tolerance and insect resistance. It seems very unlike that gene editing tools can substantially change this bias, mostly due to the polygenic control of crop performance in abiotic stress conditions. However, it is worth mentioning that innovative biotechnological tools are being developed to create cereal plants capable of fixing atmospheric N. This goal is being addressed by the Gates Foundation with support to two distinct long-term appealing approaches: the transference of bacterial genes involved in N fixation [263]; and the promotion of symbiotic interactions with N-fixing microorganisms [264]. Genetic engineering of RuBisCo and other components of the carbon fixation metabolism are also being targeted to improve plant productivity [201]. Furthermore, a collaborative project aims to develop $C_4$ rice by engineering the photosynthetic machinery of rice plants to include functional components of the $C_4$-type plant pathway (https://c4rice.com/ (accessed on accessed on 22 December 2020)). It is thus feasible that GM cereal varieties with fewer fertilizer requirements and improved photosynthetic efficiency may be released in the ongoing decade [265].

**Supplementary Materials:** The following are available online at https://www.mdpi.com/2073-439 5/11/2/376/s1, Table S1: Methodology followed for the bibliometric and bibliographic analyses.

**Author Contributions:** Conceptualization, E.B. and E.G.; methodology, E.B.; formal analysis, E.G.; writing—original draft preparation, E.B. and E.G.; writing—review and editing, E.B. and E.G. All authors have read and agreed to the published version of the manuscript.

**Funding:** The authors were funded by the Spanish Ministry of Science and Innovation (Grant No. PID2019-109089RB-C32), and by Comunidad de Madrid (Spain) and Structural EU Funds 2014-2020 (ERDF and ESF) (Grant No. AGRISOST-CM S2018/BAA-4330).

**Acknowledgments:** The authors at grateful to P. Giraldo and L. Pascual for their helpful comments to a former version of the manuscript.

**Conflicts of Interest:** The authors declare no conflict of interest.

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
