# Peer review of "Modern Approaches for the Genetic Improvement of Rice, Wheat and Maize for Abiotic Constraints-Related Traits: A Comparative Overview"

_agronomy, doi:10.3390/agronomy11020376_

Round 1

Reviewer 1 Report

The review paper “Modern Tools and Genomic Approaches for  Sustainability-oriented Genetic Improvement of  Rice, Wheat and Maize” reports an analysis at a bibliographic level about the genetic and genomic approaches utilized in the three main crops (rice, wheat and maize) to improve traits related to sustainability in particular response to abiotic stress (heat and drought) and NUE.

The general impression obtained from reading the review is that of a bibliographic inventory without specific insights.

Although the subject is of great interest, I believe that it has been treated in a somewhat superficial way without giving specific examples of important results obtained using the techniques and approaches described. There is a long list of references also very recent, and I think it would be worth to valorize within the text some of them, commenting more clearly the important achievement obtained using the different approaches.

I think the manuscript needs to be improved in order to be worthy of publication.

Specific comments

Introduction

The introduction must be enriched in bibliographic references

Line 48-49 you have forgotten the GxE interaction and the epistatic effects that may influence the expression of traits

Line 61 specify which are what you call ‘sustainability-related goals’ in the three crops. The could not be exactly the same in the three species

Line 71 [e. g., 2,3-7]  should be [2-7].

Line 59-77 I think this should be the last paragraph of the introduction describing the aim and giving an overview of the review.

Line 151 ‘Earlier marker systems had several disadvantages….’  Specify to which kind of markers you are referring.

SNPs are now the best tool, however also SSRs markers have proved to be scalable to obtain high throughput analysis.

Line 214  ‘MABC facilitates the transference of’  use ‘transfer’ instead of transference

Line 374 ‘transference RNA’ do you mean transfer RNA (tRNA)

Line 455 ‘set of gene biomarkers whose expression profiles can be a cost-effective’, do you mean a cost-effective tool?

Line 531 Genetic editing or Genome editing, not edition

Table 1,2, 3, it would be interesting to have also the references and not only the year

Figure 5 why you did not search for tilling and/ or mutagenesis

Reviewer 2 Report

E. Benavente and E. Giménez summarised the tools and methods to assist current crop breeding, which is timely and interesting. In this review, they selected rice, wheat, and maize as examples to demonstrate the successful application of the tools and methods discussed in crop improvement. After reading the manuscript, I have some concerns, which the authors may need to address.   1. The authors may describe the reasons in detail that they select rice, wheat, and maize as the examples (may give some stats to demonstrate the importance of these crops). Similarly, the authors may need to detail the reason to select drought and heat tolerance and NUE as the key traits. 2. Line 28, Page 1: The authors may briefly describe how those selected individuals have been used for crop breeding 3. Line 30, Page 1: ‘which often differ radically from their wild ancestors’. Is this true? In soybean, for example, the wild types and the cultivated lines are genetically close to each other. May reword this. 4. The authors may need to add a figure to illustrate the differences between classical breeding and modern breeding to help readers to better follow the manuscript 5. For all figures, the authors may clearly state that since which year the records have been checked 6. GWAS is a powerful tool to help identify the association between phenotype and genotype, however, it has some drawbacks. For instance, multiallelic sites are usually excluded, which may overlook the underlying function. It would be good to discuss the drawbacks of GWAS rather than only stating the advantages on page 5 7. For all tables, maybe the corresponding references are needed to support the years 8. Line 281, Page 7: does a high-quality genome assembly matter? The rice genome has been well assembled since many years ago, however, the wheat and maize assemblies are recently well assembled. I guess this may also be stated in the text  9. Line 401, Page 11: Interestingly, the NUE-related gene expression studies in rice are lacking, but to my knowledge, nitrogen is an important element for rice growth, particularly for japonica rice. Can the authors double-check this? 10. For section 4, the authors may need to discuss some regulation/policy issues related to genetic modification and give some perspectives, as in most countries, GM crops, particularly staple food crops are not allowed. The disadvantages of some methods, for instance, CRISPR, are also missing, the authors may need to add the information 11. In the last section, the authors may discuss a bit how current methods and tools can help breeding and what’s the difficulties in applying them and how can they be used to support food security particularly under the threats of the changing climate and the expanding population

Round 2

Reviewer 2 Report

Thanks authors for the update of the manuscript. I have no other questions. 

Author Response

Thanks. No remarks to respond.